# Preliminary Study: Micropropagation Using Five Types of Chelated Iron and the Subsequent Acclimation of Blue Honeysuckle (*Lonicera caerulea* var. *kamtschatica* Sevast.)

Alexey Glinushkin [1], Svetlana Akimova [1,2,*], Elena Nikulina [3], Nina Tsirulnikova [3], Vadim Kirkach [4], Valery Kalinitchenko [1,5], Agamagomed Radzhabov [2], Elena Radkevich [2], Liudmila Marchenko [2], Alexandr Solovyov [2], Alexandr Zubkov [2], Maria Panova [2], Anastasia Konstantinovich [2] and Vladimir Indolov [2]

[1]  All-Russian Phytopathology Research Institute, St. Institute, Big Vyazyomy 143050, Russia
[2]  Russian State Agrarian University, Moscow Timiryazev Agricultural Academy, Timiryazevskaya Street, 49, Moscow 127434, Russia
[3]  National Research Center "Kurchatov Institute", Square Akadimica Kurchatov, 1, Moscow 123182, Russia
[4]  Federal State Institution 'Federal Research Centre 'Fundamentals of Biotechnology' of the Russian Academy of Sciences', Leninsky Prospekt, Building 33, Building 2, Moscow 119071, Russia
[5]  Institute of Fertility of Soils of South Russia, 2, Krivoshlykova St., Persianovka 346493, Russia
*  Correspondence: akimova@rgau-msha.ru

**Abstract:** Blue honeysuckle (*Lonicera caerulea* var. *kamtschatica* Sevast.) is a valuable berry crop with a unique biochemical composition. It is rich in vitamins, minerals, and biologically active substances. Different species and cultivars of honeysuckle require different cultivation conditions in the field of accelerated reproduction in vitro. Taking into account the high clonal replication potential of the plant, we conducted research on the chelated-iron form's influence on the micropropagation productivity of the blue honeysuckle "Lulia" cultivar at the multiplication, rooting, and subsequent acclimatization stages of microplants. In a preliminary study, five types of iron chelates were tested with carboxyl- and phosphorus-containing ligands: Fe(III)-EDTA, Fe(III)-DTPA, Fe(III)-EDDHA, Fe(III)-HEDP, and Fe(II)-HEDP. Each type of iron chelate was applied at four concentrations: standard, decreased by 2 times, and increased by 1.5 times and 2 times in the basic Murashige and Skoog (MS) nutrient medium. It was found that the blue honeysuckle "Lulia" had a selectivity to the type of iron chelate that was used. The nutrient-medium modifications with iron chelates, which caused the plant response, contributed to a significant improvement in the plant's physiological status and increased its survival rate during the microplant's acclimation to the nonsterile conditions stage. At the rooting stage, an increase in the rooting rate of up to 100% (Fe (III)-EDDHA), an increase in the number of shoots by 1.5–2 times, and an increase in the number of roots by 1.4–1.9 times were observed. The positive effect of the iron's chelated forms was also observed at the acclimation stage. According to the results of the research, the most suitable iron forms for clonal micropropagation of blue honeysuckle were carboxyl-containing Fe(III)-EDDHA and phosphonate-containing Fe(II)-HEDP. Moreover, the effectiveness of both complexonates was confirmed in a wide concentration range: Fe(III)-EDDHA from ($\times$1.0) to ($\times$2.0), and Fe(II)-HEDP from ($\times$0.5) up to ($\times$1.5).

**Keywords:** chelate; hydroxyethylidene diphosphonic acid; culture medium; proliferation; rooting; in vitro





## 1. Introduction

Currently, the World Health Organization is concerned about the modification and changes of the nutrition of populations, in particular, the lack of dietary fiber, vitamins, and minerals in people's diets [1,2]. Therefore, fruit and vegetables have a great importance in a healthy diet. They are natural sources of biologically active substances and have high taste qualities and therapeutic and prophylactic properties; that is why the demand for them is constantly growing [3,4].

Blue honeysuckle (*Lonicera caerulea* var. *kamtschatica* Sevast.) is a berry crop with a unique biochemical composition. It contains a lot of vitamins and biologically active substances with medicinal properties which are accepted as official medicine [5,6].

Honeysuckle is especially valuable in the climatic zones of Russia [6,7]. The early summer period of ripening allows for populations to supplement their diet with fresh berries during the period of "vitamin hunger". This crop is characterized by a high winter hardiness and resistance to spring frosts and fungal diseases. Honeysuckle favors modest growing conditions. It is a good early honey plant, which is also suitable for mechanized harvesting [5,7]. The plant's advantages have contributed to a development of honeysuckle production in the USA, Canada [6], in European countries (Belarus, Croatia, Czechia, Poland, Romania, Slovakia, Ukraine) [8–17], and China [18,19].

Nowadays, the technologies of intensive honeysuckle cultivation aim to increase production and aim to find the best ways to obtain high-quality planting material in large quantities. [20]. That is why many studies are devoted to effective methods for accelerated honeysuckle cultivation in vitro [8,9,13,16,17]. According to published studies, there are certain difficulties in creating a universal method, because the composition of the nutrient medium and the explant reproduction conditions of the tested plants depend on their genotype [10,19,20].

There are a lot of studies devoted to honeysuckle micropropagation in "in vitro" culture. Applying hormonal growth regulators at different concentrations on the basis of standard nutrient media, such as Murashige and Skoog's (MS) medium [21], Driver and Kuniyuki's medium (DKW) [22] and Woody Plant Medium (WPM), are the main themes of such studies. Thus, at the proliferation stage, the following substances were added to the nutrient medium: 6-benzylaminopurine (BAP) [10,13,14,23] or different combinations of plant growth regulators (PGRs) (6-benzylaminopurine (BAP), thidiazuron (TDZ), and kinetin (KIN)) with different auxins (indole-3-butyric acid (IBA) and indole-3-acetic acid (IAA)) [12]. Scientists have researched the effectiveness of adding auxins: indole-3-butyric acid (IBA), indole-3-acetic acid (IAA), and 1-naphthaleneacetic acid (NAA) [13,15,24].

It should be noted that a focus on only the selection of hormonal components limits the scope of searching for the optimal composition of a nutrient medium, since other vital plant nutrients, such as iron, for example, remain outside of the researchers' attention. Earlier research by various authors noted the exceptional importance of iron in crop propagation in vitro [25–28]. It has been shown for many crops that replacing the standard $FeSO_4$ component together with the disubstituted sodium salt of ethylenediaminetetraacetic acid ($Na_2EDTA$) directly with the FeEDDHA complex leads to improved iron nutrition [28–30]. Among the published studies, there is information about the influence of iron compound replacement with the FeEDDHA complex on the growth and development in vitro of various crops: papaya, blueberry, raspberry, hazelnut, olive, baptisia, peach, etc. [31–36]. At the same time, in most studies, the focus has been the question of replacement with one of the most popular FeEDDHA iron complexes. In our previous research, we determined that the valence of element and the nature of ligands had a great influence on the development of explants as much as the direct replacement of iron in the nutrient medium with the chelated form [37,38]. Iron complexes with both known carboxyl-containing ligands (EDTA, DTPA, EDDHA) and phosphorus-containing ligands (HEDP) were tested in the study. It should be noted that a study on the use of metal complexonates with phosphorus-containing ligands, including iron, in nutrient media for plant clonal micropropagation has not been conducted before. In our research, we aimed to find out the features and opportunities of phosphorus-containing complexonates by applying them in in vitro culture. Organophosphorus complexons are able to form, under certain conditions, stable complexes both with Fe(III) and Fe(II), which is impossible in relation to carboxyl ligands, while the divalent form of Fe is more physiologically preferable for the vital activity of many plants.

Phosphorus-containing complexons contain phosphonic groups in their structure, which, at certain concentrations, can have a regulatory effect on the growth and development of plants and promote resistance to stress.

At all stages of in vitro and ex vitro honeysuckle growth, the environmental conditions present determine the development of plants. It is important to overcome the limitations of plant development. The potential of a substrate water matrix under standard in vitro and ex vitro conditions is excessively high. A plant's acclimation to ex vitro conditions at high humidity leads to the plant's addiction to a high soil moisture rate, which deteriorates acclimatization in the field. A high substrate humidity causes a free penetration of heavy metals from the soil into the plant and reduces the quality of the product. The same effect is observed when watering plants using standard irrigation. In order to reduce the entry of heavy metals into plants, a method of intrasoil-pulsed continuous–discrete moistening, which ensures the passivation of heavy metals, has been developed [39].

Taking into account the high reproduction potential of honeysuckle for a healthy diet and, consequently, the growing interest in the mass production of this berry crop, in this research, we aimed to study in detail the influence of different forms of iron at the micropropagation stages of blue honeysuckle "Lulia" (*Lonicera caerulea* var. *kamtschatica* Sevast.). It seemed important to us to investigate each stage of the clonal micropropagation cycle separately: proliferation, rooting, subsequent acclimatization, and cultivation. Thus, we aimed to determine the effectiveness of this technique for the entire technological cycle of honeysuckle cultivation from the perspective of ensuring a successful in vitro plant production.

## 2. Materials and Methods

### 2.1. Plant Material and Chemicals

Studies were carried out on the blue honeysuckle "Lulia" (*Lonicera caerulea* var. *kamtschatica* Sevast.). The cultivar was obtained as a result of the free pollination of blue honeysuckle plants. The bush is of medium height with large, juicy, sweet berries midseason.

The experiments were carried out in 2020–2021 at the Russian State Agrarian University, Moscow Timiryazev Agricultural Academy, in the department of biotechnology and berry crops of the Edelstein Educational Scientific and Production Center for Horticulture and Vegetable Growing. Analytics and forecasting were performed at the State Scientific Institution "All-Russian Research Institute of Phytopathology" in the Department of Resistance Science.

Samples of aqueous solutions of iron complexonates for modifying nutrient media Fe(III)-EDTA (salt of ethylenediaminetetraacetic acid), Fe(III)-DTPA (salt of diethylenetriamine pentanoic acid), Fe(III)-EDDHA (salt of ethylenediamin-N,N'-o-oxyphenyl-N, N'-dioacetic acid), Fe(III)-HEDP (hy-droxyethylidine diphosphonic acid salt), and Fe(II)-HEDP were provided by the Laboratory of Technology of Complexons and Complex Compounds of the Kurchatov Institute. Analytical research of the reagent samples was performed using the scientific equipment of the Kurchatov Institute Research Center.

Nutrient Medium and Growth Conditions

For the establishment of the in vitro meristems, nodal segments of blue honeysuckle "Lulia" were collected from ten donor plants and washed with running water for 45 min. They were then immersed in a fungicidal solution containing $0.2$ mL L$^{-1}$ of QUADRIS® (22.9% azoxystrobin:methyl (E)-2-{2-[6-(2-cyanophenoxy) pyrimidin-4-yloxy] phenyl}-3-methoxyacrylate) for 30 min.

Within a horizontal laminar flow chamber, they were washed in 70% alcohol solution for 1–2 s. They were then immersed in a 1.5% NaOCl solution and 0.1% Tween 20 for 15 min with constant shaking. Finally, the nodal segments were washed in triple-autoclaved water for 5 min.

Then, the meristems were dissected under a stereo microscope ($1.6\times$ or $2.0\times$, Carl Zeiss Microscopy STEMI 2000).

The isolated meristems were inoculated vertically and placed into 15 cm $\times$ 2.5 cm test tubes containing 10 mL of culture medium with mineral salts according to Murashige and Skoog (MS) [24], enriched with the following substances: thiamine hydrochloride (B1), pyri-

doxine hydrochloride (B6), nicotinamide (PP), 0.5 mg L$^{-1}$; glycine, 1 mg L$^{-1}$; mesoinositol, 100 mg L$^{-1}$; 6-benzylaminopurine (6-BAP), 0.2 mg L$^{-1}$; sucrose, 30,000 mg L$^{-1}$; and plant micropropagation agar-agar (DIA-M, Russia) (gel strength > 1000 g/cm$^2$ turbidity < 50 NTU, pH (1.5%) 5.80), 6000 mg L$^{-1}$. Once established, the plants differentiated from the meristems were cultivated in a 200 mL glass.

At the stage of multiplication, explants of blue honeysuckle "Lulia" (stage S1) were planted on a nutrient medium with mineral salts according to Murashige and Skoog (MS) [21], enriched with the following substances: thiamine hydrochloride (B1), pyridoxine hydrochloride (B6), nicotinamide (PP), 0.5 mg L$^{-1}$; glycine, 1 mg L$^{-1}$, mesoinositol, 100 mg L$^{-1}$, 6-benzylaminopurine (6-BAP), 0.2 mg L$^{-1}$; sucrose, 30,000 mg L$^{-1}$; and plant micropropagation agar-agar (DIA-M, Russia) (gel strength > 1000 g/cm$^2$ turbidity < 50 NTU, pH (1.5%) 5.80), 6000 mg L$^{-1}$.

At the stage of rooting (stage S2), the explants were placed on a nutrient medium with mineral salts according to the Murashige and Skoog's (MS) recipe [24] for the stage of rooting, which contained $\frac{1}{2}$ part of macroelements and microelements enriched with the following substances: thiamine hydrochloride (B1), pyridoxine hydrochloride (B6), nicotinamide (PP), 0.5 mg L$^{-1}$; indole-3-butyric acid (IBA), 0.5 mg L$^{-1}$; sucrose, 15,000 mg L$^{-1}$; and agar-agar, 6000 mg L$^{-1}$.

In total, 20 modifications of the Murashige and Skoog (MS) nutrient medium were involved in the experiment series at the stage of root formation: 5 types of iron chelates in 4 concentrations (Table 1).

**Table 1.** Experimental concentrations of iron chelates in the modified nutrient media.

| Complex | Molar Mass, g | Iron Content in Powder/in Solution, % | Concentration of Fe Solution, mgL$^{-1}$ | | | |
|---|---|---|---|---|---|---|
| | | | Reduced (×0.5) Fe-2.79 | Standard (×1.0) Fe-5.58 | Increased (×1.5) Fe-8.37 | Doubled (×2.0) Fe-11.16 |
| Fe(III)-EDDHA (mL L$^{-1}$) | 439 | In solution 0.015 | 18.6 | 37.2 | 55.8 | 74.4 |
| Fe(III)-EDTA (mg L$^{-1}$) | 368 | In powder 13.4 | 0.02 | 0.04 | 0.06 | 0.08 |
| Fe(III)-DTPA (mg L$^{-1}$) | 470 | In powder 9.81 | 0.03 | 0.06 | 0.09 | 0.12 |
| Fe(III)-HEDP (mL L$^{-1}$) | 259 | In solution 0.65 | 0.43 | 0.86 | 1.29 | 1.72 |
| Fe(II)-HEDP (mL L$^{-1}$) | 260 | In solution 0.52 | 0.54 | 1.07 | 1.61 | 2.15 |

Instead of the traditionally used FeSO$_4$ × 7H$_2$O + Na$_2$EDTA (control), five iron complexons were introduced into the composition of the nutrient medium: Fe(III)-EDTA is a complex of iron with ethylenediaminetetraacetic acid (Fe content in the powder = 13.4%); Fe(III)-DTPA is an iron complex with diethylenetriaminepentaacetic acid (Fe content in powder = 9.81%); Fe(III)-EDDHA is an iron complex with ethylenediamine-bis-hydroxyphenylacetic acid (Fe content in solution = 0.015%); Fe(II)-HEDP is a complex of iron from the bisphosphonate class with exciethylidene diphosic acid (Fe content in solution = 0.52%); and Fe(III)-HEDP is a complex of iron with hydroxyethylidene diphosphonic acid (Fe in solution = 0.65%).

In the nutrient medium prescribed by Murashige and Skoog, FeSO$_4$ × 7H$_2$O is used in a quantity equal to 27.8 mg L$^{-1}$. The molar mass of FeSO$_4$ × 7H$_2$O is 278.05 g, in which the molar mass of iron (Fe) is 55.85 g, indicating that 27.8 mg of FeSO$_4$ × 7H$_2$O contains 5.58 mg of iron. The amount of Fe in the experimental variants was calculated depending on the initial amount in the MS recipe, 5.58 mg, and the following concentrations were used: ×0.5—half reduced (2.79 mg L$^{-1}$), ×1.0—standard (5.58 mg L$^{-1}$), ×1.5—increased by 1.5 times (8.37 mg L$^{-1}$), and ×2.0—doubled (11.16 mg L$^{-1}$).

The pH of the medium was adjusted to 5.8 before autoclaving and the medium was poured into glass culture vessels with a volume of 200 mL, while each vessel was filled with 30 mL of the medium. Further vessels with the nutrient medium were sterilized using pressurized saturated steam ($p$ = 101 kPa) at 120 °C for 20 min.

The process of planting the explants on a medium was carried out in a laminar box according to the rules for working with sterile materials.

The explants for the research were obtained as follows: the part of the apical shoot of the mother plant was taken and divided into several parts. At the propagation stage, parts of microshoots containing 2–3 phytomers (15–20 mm long) were used as explants, and 5 pieces were planted in each vessel. At the rooting stage, parts of the microshoots containing 2–3 phytomers (15–20 mm long) were used as explants, and 15 pieces were planted in each culture vessel.

At the multiplication and rooting stage, the plants were maintained for 60 days in a growth room under fluorescent lamps (46.25 $\mu Mm^{-2}s^{-1}$ PPFD) and a 16/8 h (light/dark) photoperiod, at $20 \pm 2$ °C. Observations and measurements were recorded afterward and at 60 days of culture.

At the end of the rooting stage, the transplant-ready microplants were thoroughly washed of the nutrient medium and planted for acclimation (stage S3) to nonsterile conditions in plastic cassettes with a cell diameter of 4 cm, which were preliminarily filled with a mixture of enriched peat "Pelgorskoe-M" with perlite in the ratio of 3:1. The peat nutrient substrate contained nitrogen ($NH_4 + NO_3$) at 100 mg $L^{-1}$, phosphorus ($P_2O_5$) at 90 mg $L^{-1}$, and potassium ($K_2O$) at 120 mg $L^{-1}$; the pH of the salt suspension was no less than 5.5 and the mass fraction of moisture was 65%. Exactly 24 h before planting the microplants, the substrate was saturated with water and then shed with a solution of the fungicide Previcur at a concentration of 2 mL $L^{-1}$. The honeysuckle microplant cassettes were placed in a greenhouse, where the air humidity was maintained for 2 weeks at a level of 80%–90%, under a natural photoperiod and irradiance with day/night at $20 \pm 5$ °C$/18 \pm 5$ °C. During the acclimation period, beginning from the 14th day after the transfer to ex vitro conditions, the humidity was reduced step by step by daily ventilation of the greenhouse. On the 50th day of the acclimation stage, the parameters of the microplants' growth and development were recorded.

### 2.2. Morphological Measurements and Observations

The morphometric and biometric parameters of the blue honeysuckle microshoots were recorded according to the list of indicators presented in Table 2.

**Table 2.** List of the measured morphometric indicators of plants.

| Parameter | Multiplication | Rooting | Acclimation |
|---|---|---|---|
| Multiplication factor | + | | |
| Rooting rate, % | | + | |
| Survival rate, % | | | + |
| Average number of shoots, pcs. | | + | |
| Average shoots length, cm | + | + | + |
| Leaf surface area, cm$^2$ | | | + |
| Average number of roots, pcs. | | + | + |
| Average roots length, cm | | + | + |
| The total length of the roots, cm | | + | + |

### 2.3. Statistical Analysis

In a preliminary study, the experiments were performed in triplicate with 5 plants per repetition (a total of 15 plants). Statistical processing of the results was carried out according to A.V. Isachkin's method [40] and using STATISTICA_10.0.1011. When determining the significant difference between the experimental variants, the least significant difference (LSD) at a level of $p < 0.05$ was used.

## 3. Results

### 3.1. The Effect of Nutrient Medium Modification on the Multiplication Ability of the Blue Honeysuckle Explants (Stage S1)

Applying different iron complexes to the nutrient medium caused various explant reactions to occur in the proliferation process. At the same time, both the type of complex,

and its concentration influenced the microplant growth. Figures 1 and 2 demonstrate the shoot multiplication factor and the average shoot length diagrams observed during the experiment.

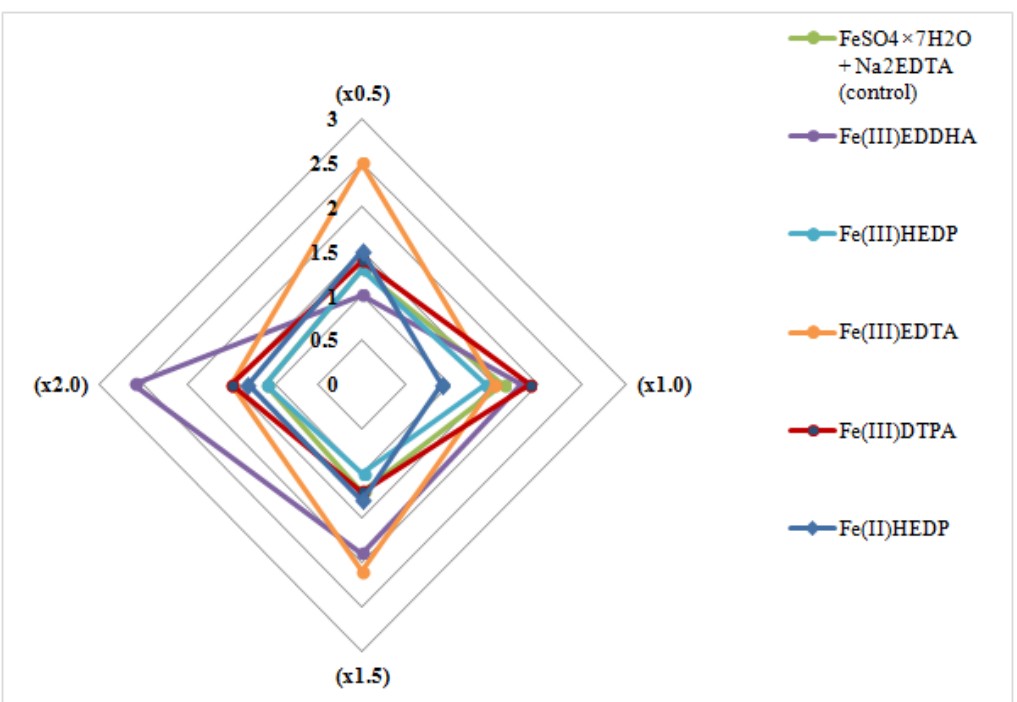

**Figure 1.** Shoot multiplication factor of blue honeysuckle "Lulia" (*Lonicera caerulea* var. *kamtschatica* Sevast.) when applying different sources of iron in 4 concentrations: reduced (×0.5), standard (×1.0), increased (×1.5), and doubled (×2.0).

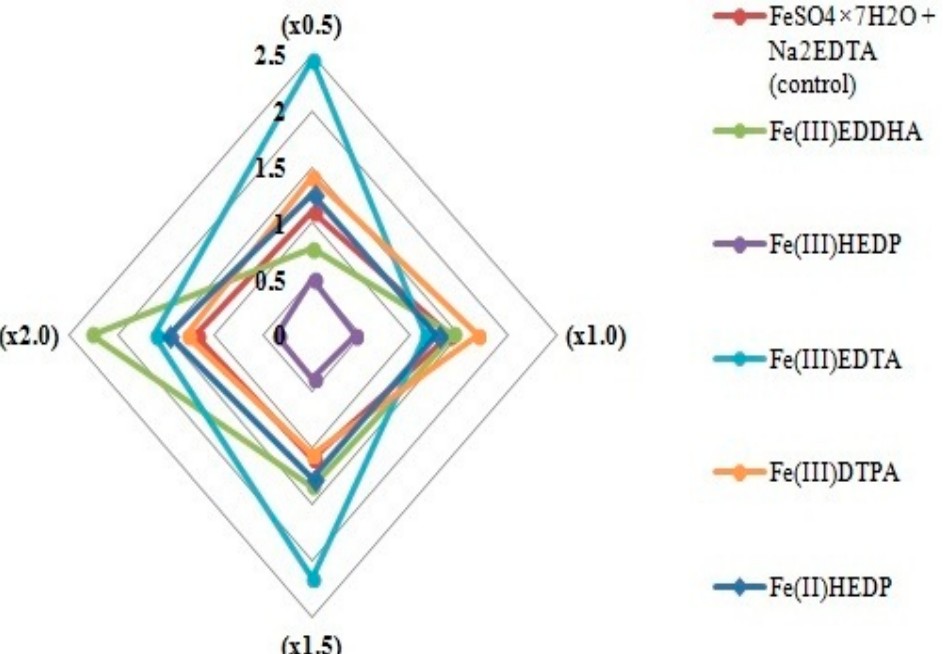

**Figure 2.** Average shoot length (cm) of blue honeysuckle "Lulia" (*Lonicera caerulea* var. *kamtschatica* Sevast.) when applying different sources of iron in 4 concentrations: reduced (×0.5), standard (×1.0), increased (×1.5), and doubled (×2.0).

The chelates with carboxyl-containing ligands provided a high increase in the shoot multiplication factor of the microplants compared with the control variants when the concentration range increased from 1.5 to 2.6, which was 1.6 and 2.4 higher than in the control variants. In that case, Fe-EDTA had the maximum versatility, but the Fe-EDDHA and Fe-DTPA complexes also promoted a significant increase in the shoot multiplication factor in concentrations from standard to double. Bisphosphonate based on iron chelates showed less efficiency. The values of the shoot multiplication factor were similar to those of the control and, in some cases, slightly lower. This was notably observed for the trivalent iron complex Fe(III)-HEDP (1.4 when the standard concentration was applied and 1.1 when the double concentration was applied).

A similar pattern was observed with the average shoot length index (Figure 2). On the nutrient media with the addition of carboxyl complexes, the explants increased their growth rate in different proportion from 1.1 to 1.9 times. In addition, best results were also observed in variants with Fe-EDTA and Fe-EDDHA applied. The growth inhibition of the explants cultivated on the bisphosphonate-based iron chelate medium with the trivalent form of iron (Fe(III)-HEDP) was observed at all experimental concentration variants. The average shoot length was decreased by threefold compared to the control variants, whereas the microplants cultivated on the medium based on Fe(II)-HEDP had similar growth rates to those of the control variants. The double concentration treatment increased the shoot length by 25%. It should be noted that the application of iron chelates with organophosphorus ligands in the nutrient medium caused physiological reactions such as leaf necrosis and the chlorosis of the plants (no more than 10%).

### 3.2. The Effect of Nutrient Medium Modification on the Rooting Rate of Blue Honeysuckle Explants during the Rooting Stage (Stage S2)

The plant's response tendency to various iron forms and their concentrations had already been observed after the intermediate monitoring of microplant growth on the 45th day of the experiment. On the 60th day of cultivation, the physiological response of the microshoots was more pronounced. The recorded indicators of growth and development of honeysuckle microgreens at the rooting stage at the 60th day of cultivation are presented in Table 3.

Despite the very good ability of this culture to spontaneously rooting on standard nutrient medium, the introduction of iron-chelated forms had an additional stimulating effect of up to 100% rooting for all the shoots (Figure 3A). The highest rooting rate (100%) of blue honeysuckle "Lulia" microshoots was observed in the medium with the Fe(III)-EDDHA complex in three concentrations; the standard increased by 1.5 and 2 times. Different iron compounds had different effects on the development of both the above-ground organs (shoots) and the root system's structure.

The presence of Fe(III)-EDDHA at an increased concentration ($\times$1.5) and Fe(III)-DTPA at all concentrations promoted shoot explant development at a rate of 1.5–2.0 times more than in the other variants of the experiment. A significant improvement in root system structure development (the number of roots and the average root length) was observed throughout the entire concentration range of carboxyl-containing Fe(III)-EDDHA and Fe(III)-EDTA iron chelates and Fe(II)-HEDP bisphosphonate. Thus, in the presence of Fe(III)-EDDHA, the plants developed about 1.5 times more roots compared to the control variants; in the presence of Fe(III)-EDTA, this was 1.7–1.9 times more and for Fe(II)-HEDP, this was 1.4–1.7 times more. At the same time, in the variants with carboxyl-containing Fe(III)-EDDHA and Fe(III)-EDTA ligands, not only an increase in the number of shoots but also an increase in their length (by 40%–80%) was observed (Table 3, Figure 4).

**Table 3.** The effect of the introduction of chelated iron compounds into the nutrient medium on the development of blue honeysuckle "Lulia" (*Lonicera caerulea* var. *kamtschatica* Sevast.) at the stage of rooting.

| Concentrations of Iron Chelates in Modified Nutrient Media (Factor A) | Iron Chelates (Factor B) | | | | | | Factor A Mean |
|---|---|---|---|---|---|---|---|
| | FeSO$_4$ × 7H$_2$O + Na$_2$EDTA + SD * (Control) | Fe(III)-EDDHA + SD | Fe(III)-EDTA + SD | Fe(III)-DTPA + SD | Fe(III)-HEDP + SD | Fe(II)-HEDP + SD | |
| | Rooting rate, % | | | | | | LSD$_{05}$ a = 7.49 |
| ×0.5 | 85.0 ± 3.03 | 80.0 ± 2.47 [a**] | 88.0 ± 4.56 | 87.0 ± 2.77 | 47.0 ± 5.55 | 87.0 ± 2.25 | 79.0 |
| ×1.0 | 87.0 ± 2.52 | 60.0 ± 6.63 | 87.0 ± 2.33 | 100 ± 0.00 [b] | 0 | 87.0 ± 3.76 | 70.2 |
| ×1.5 | 88.0 ± 3.10 | 86.0 ± 0.47 [a] | 93.0 ± 2.33 | 100 ± 0.00 [b] | 0 | 67.0 ± 5.57 | 72.3 |
| ×2.0 | 87.0 ± 3.62 | 60.0 ± 1.36 | 67.0 ± 6.25 | 100 ± 0.00 [b] | 0 | 67.0 ± 4.65 | 63.5 |
| Factor B mean LSD$_{05}$ b = 8.41 | 86.8 | 71.5 | 83.8 | 96.8 | 11.8 | 77.0 | |
| | LSD$_{05}$ ab = 10.40 to compare private averages | | | | | | |
| | Average number of shoots, pcs. | | | | | | LSD$_{05}$ a = 0.40 |
| ×0.5 | 1.4 ± 0.83 | 1.3 ± 0.44 | 1.4 ± 0.49 | 3.1 ± 0.44 [a,b] | 1.8 ± 0.83 [a,b] | 0.9 ± 1.36 | 1.7 |
| ×1.0 | 1.5 ± 0.72 | 1.7 ± 0.50 | 1.5 ± 0.50 | 2.5 ± 0.60 [b] | 0.9 ± 0.68 | 1.1 ± 1.36 | 1.5 |
| ×1.5 | 1.5 ± 0.79 | 3.3 ± 0.79 [a,b,ab] | 1.3 ± 0.47 | 2.5 ± 1.12 [a,b] | 1.1 ± 0.85 | 1.7 ± 1.67 [a] | 1.9 |
| ×2.0 | 1.5 ± 0.81 | 1.3 ± 0.71 | 1.7 ± 0.70 | 2.2 ± 0.44 [b] | 1.2 ± 0.83 | 1.6 ± 1.47 [a] | 1.6 |
| Factor B mean LSD$_{05}$ b = 0.34 | 1.5 | 1.9 | 1.5 | 2.6 | 1.3 | 1.3 | |
| | LSD$_{05}$ ab = 1.14 to compare private averages | | | | | | |
| | Average shoot length, cm | | | | | | LSD$_{05}$ a = 0.39 |
| ×0.5 | 1.6 ± 0.85 | 2.4 ± 0.90 [a,b] | 2.2 ± 1.38 [a,b] | 1.0 ± 1.20 | 0.9 ± 0.39 | 0.9 ± 0.43 | 1.5 |
| ×1.0 | 1.8 ± 0.72 | 2.1 ± 1.49 [b] | 2.1 ± 1.09 [b] | 0.9 ± 1.01 | 0.4 ± 0.34 | 1.7 ± 0.36 | 1.5 |
| ×1.5 | 1.5 ± 0.84 | 0.8 ± 0.98 | 1.8 ± 0.94 [b] | 0.9 ± 0.50 | 0.4 ± 0.40 | 1.4 ± 0.44 | 1.1 |
| ×2.0 | 1.4 ± 0.95 | 1.5 ± 1.08 | 1.3 ± 0.85 | 0.8 ± 0.63 | 0.6 ± 0.47 | 1.5 ± 0.30 | 1.2 |
| Factor B mean LSD$_{05}$ b = 0.33 | 1.6 | 1.7 | 1.9 | 0.9 | 0.6 | 1.4 | |
| | LSD$_{05}$ ab = 1.13 to compare private averages | | | | | | |
| | Average number of roots, pcs. | | | | | | LSD$_{05}$ a = 0.97 |
| ×0.5 | 2.7 ± 1.59 | 3.7 ± 2.60 [b] | 4.2 ± 2.48 [b] | 3.1 ± 2.24 | 1.1 ± 1.54 [a] | 3.6 ± 2.38 [b] | 3.1 |
| ×1.0 | 2.5 ± 1.45 | 3.9 ± 3.07 [b] | 4.4 ± 2.55 [b] | 3.3 ± 1.77 [b] | 0 | 3.1 ± 1.89 | 2.9 |
| ×1.5 | 2.6 ± 1.77 | 3.3 ± 3.19 | 5.5 ± 2.60 [a b] | 2.3 ± 1.12 | 0 | 4.9 ± 2.30 [a,b] | 3.1 |
| ×2.0 | 2.9 ± 1.91 | 4.4 ± 2.06 [b] | 4.2 ± 3.56 [b] | 3.0 ± 1.74 | 0 | 2.1 ± 2.76 | 2.8 |
| Factor B mean LSD$_{05}$ b = 0.81 | 2.7 | 3.8 | 4.6 | 2.9 | 0.3 | 3.4 | |
| | LSD$_{05}$ ab = 2.77 to compare private averages | | | | | | |
| | Average root length, cm | | | | | | LSD$_{05}$ a = 0.49 |
| ×0.5 | 1.1 ± 0.69 | 1.3 ± 0.62 | 1.7 ± 0.86 [b] | 0.7 ± 0.77 | 0.4 ± 0.53 | 1.0 ± 0.39 | 1.0 |
| ×1.0 | 1.2 ± 0.70 | 1.7 ± 0.83 [b] | 1.4 ± 0.71 | 0.7 ± 0.96 | 0 | 0.9 ± 0.38 | 1.0 |
| ×1.5 | 0.9 ± 0.68 | 1.3 ± 4.50 | 1.6 ± 0.67 [b] | 0.5 ± 0.97 | 0 | 2.2 ± 0.43 [a,b] | 1.1 |
| ×2.0 | 0.9 ± 0.69 | 1.6 ± 0.48 [b] | 1.2 ± 0.95 | 0.4 ± 0.50 | 0 | 0.5 ± 0.44 | 0.8 |
| Factor B mean LSD$_{05}$ b = 0.41 | 1.0 | 1.5 | 1.5 | 0.6 | 0.1 | 1.2 | |
| | LSD$_{05}$ ab = 1.39 to compare private averages | | | | | | |
| | Average total root length, cm | | | | | | LSD$_{05}$ a = 1.57 |
| ×0.5 | 3.3 ± 2.69 | 6.2 ± 3.51 [b] | 7.9 ± 4.56 [b] | 2.5 ± 5.19 | 1.1 ± 1.55 | 4.5 ± 2.11 | 4.3 |
| ×1.0 | 3.5 ± 2.37 | 6.7 ± 4.43 | 6.9 ± 3.83 | 2.6 ± 4.24 | 0 | 4.2 ± 1.80 | 5.9 |
| ×1.5 | 3.4 ± 2.70 | 3.9 ± 5.26 | 8.9 ± 5.44 [a,b] | 1.9 ± 2.04 | 0 | 6.1 ± 2.42 [a,b] | 4.0 |
| ×2.0 | 3.3 ± 3.01 | 7.0 ± 1.86 [b] | 7.9 ± 7.99 [b] | 1.7 ± 2.93 | 0 | 1.7 ± 1.66 | 3.6 |
| Factor B mean LSD$_{05}$ b = 1.33 | 3.4 | 6.9 | 7.9 | 2.2 | 0.3 | 4.1 | |
| | LSD$_{05}$ ab = 4.51 to compare private averages | | | | | | |

The least significant difference (LSD) $p < 0.05$ was calculated by a two-way analysis of variance. * The results are expressed as mean ± standard deviation (SD). ** "a, b, ab" are the differences between the averages when the recording is significant based on the comparison of the differences between the mean with an LSD at a 5% significance level: "a" by factor "a" (experimental concentrations of iron chelates), "b" by factor "b" (iron chelates), and "ab" by the interaction of factors.

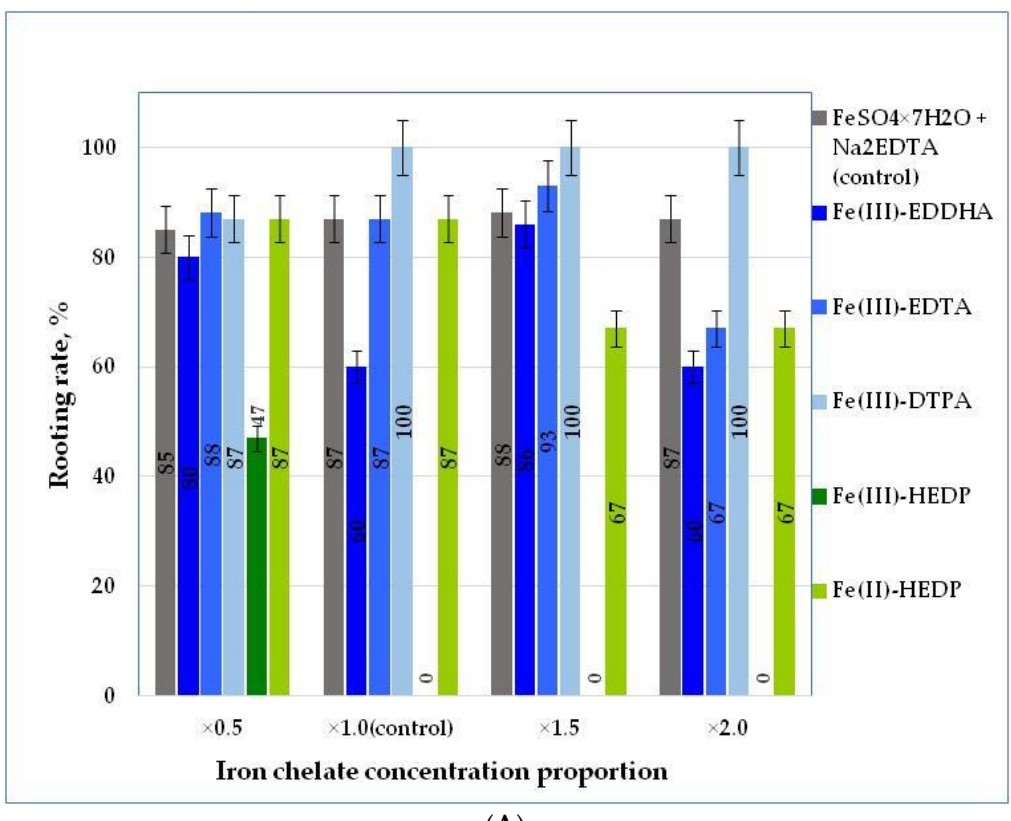

(**A**)

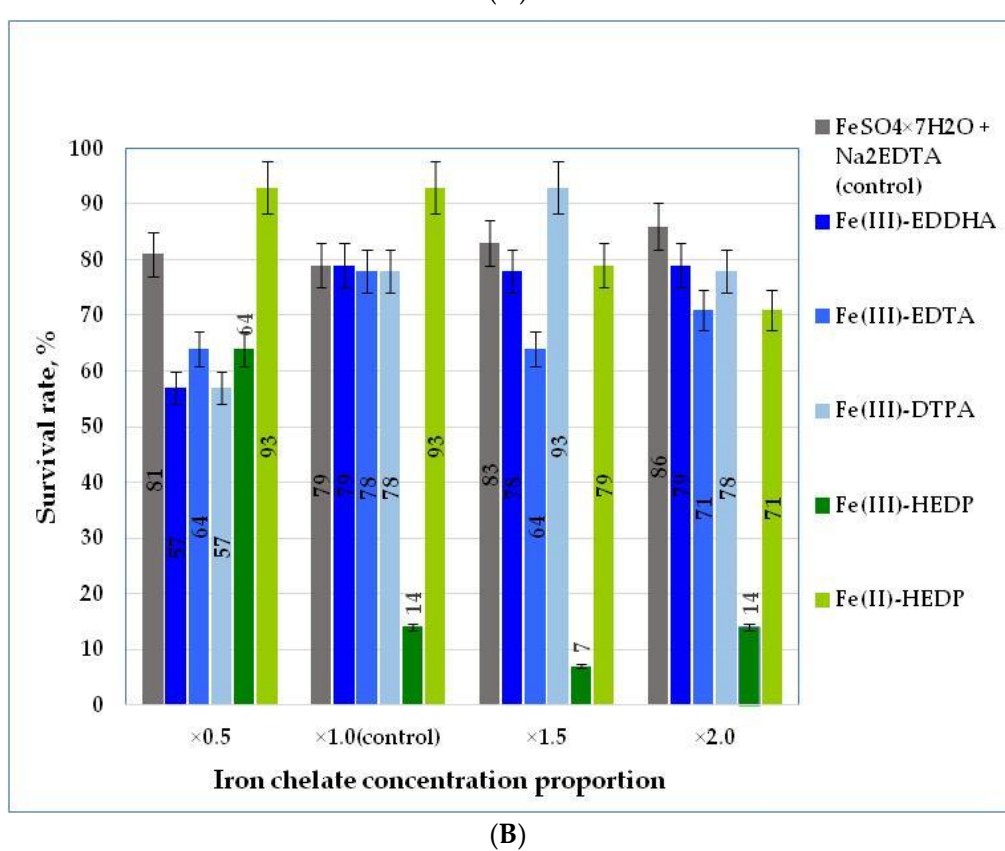

(**B**)

**Figure 3.** Rooting rate at the stage of rooting (**A**) and microplants' survival rate at the acclimatization stage (**B**) of blue honeysuckle "Lulia" (*Lonicera caerulea* var. *kamtschatica* Sevast.) depending on the iron chelate type in the MS nutrient medium.

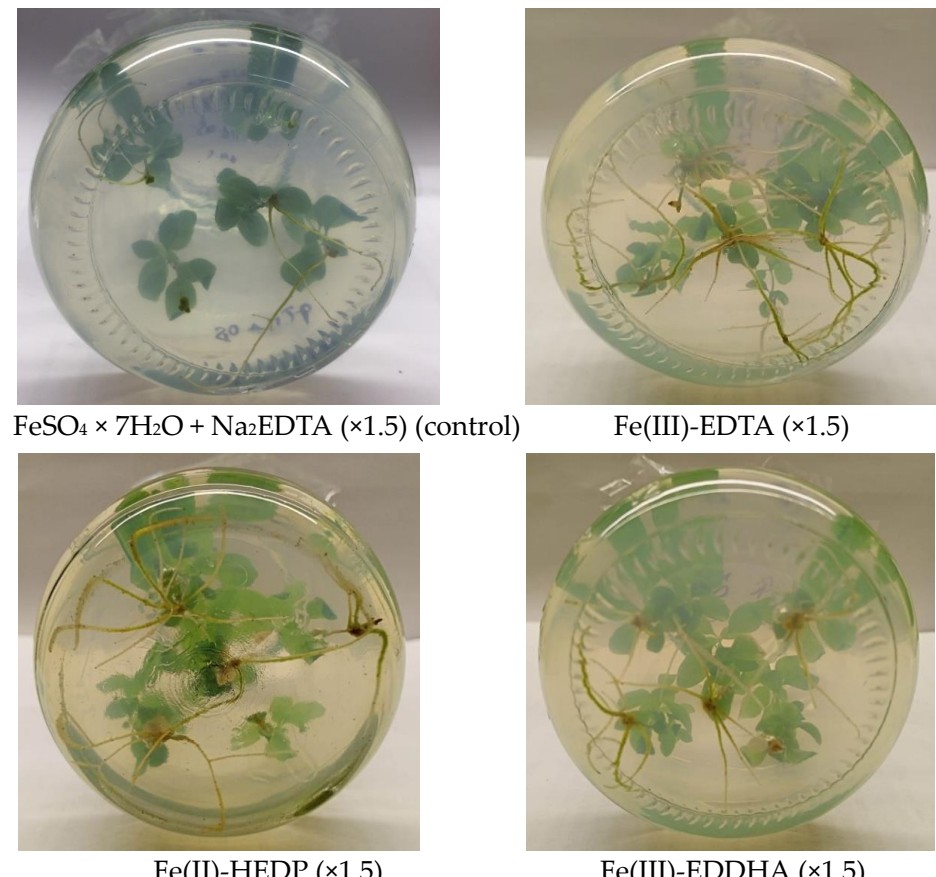

FeSO₄ × 7H₂O + Na₂EDTA (×1.5) (control)    Fe(III)-EDTA (×1.5)

Fe(II)-HEDP (×1.5)    Fe(III)-EDDHA (×1.5)

**Figure 4.** Rooting rate at the end at the stage of rooting microplants of blue honeysuckle "Lulia" (*Lonicera caerulea* var. *kamtschatica* Sevast.) depending on the iron chelate type at an increased concentration (×1.5) in the MS nutrient medium.

Special attention should be paid to the fact, that the application of iron with different ligands may cause different effects on plant growth. The application of Fe(III)-DTPA stimulated the development of above-ground organs (shoots) and some inhibition of the root system's growth. At the same time, the application of Fe(III)-EDTA and Fe(II)-HEDP led to the opposite effect, a significant strengthening of the root development and some inhibition of the shoot growth compared with the control variants. As for the nutrient medium with trivalent iron (Fe(III)-HEDP) bisphosphonate, a significant inhibitory effect on all indicators of explant development was observed. At Fe(III)-HEDP concentrations higher than (×0.5), rooting did not take place over the period of 60 days.

*3.3. Acclimation to Nonsterile Conditions of Blue Honeysuckle Microplants Rooted on Modified Nutrient Media (Stage S3)*

The results of the data analysis of the development of blue honeysuckle plants after the acclimation stage allowed us to estimate the after-effect of different chelated iron form applications in nutrient media at the rooting stage (Table 4).

As can be seen from the above data, the modification of the nutrient media with chelated forms of iron had quite a significant effect on the plant status and the number of adapted plants. All iron complexes except Fe(III)-HEDP practically in the whole range of the studied concentrations resulted in an increase of the growth and development indicators: shoot length, total leaf surface area, and the number of roots and their length. Figure 5 demonstrates blue honeysuckle microplants grown on nutrient media with Fe(III)-EDTA, Fe(III)-EDDHA, Fe(II)-HEDP, and the control variant at an increased concentration (×1.5).

**Table 4.** The influence of the type of iron and its concentration in the nutrient medium on the morphometric parameters of blue honeysuckle "Lulia" (*Lonicera caerulea* var. *kamtschatica* Sevast.) at the stage of growing plants in a minigreenhouse.

| Concentrations of Iron Chelates in Modified Nutrient Media (Factor A) | Iron Chelates (Factor B) | | | | | | Factor A Mean |
|---|---|---|---|---|---|---|---|
| | $FeSO_4 \times 7H_2O$ + $Na_2EDTA$ + SD * (Control) | Fe(III)-EDDHA+ SD | Fe(III)-EDTA + SD | Fe(III)-DTPA + SD | Fe(III)-HEDP + SD | Fe(II)-HEDP + SD | |
| | Survival rate, % | | | | | | $LSD_{05}$ a = 6.32 |
| ×0.5 | 81.0 ± 5.89 | 57.0 ± 4.42 | 64.0 ± 3.45 | 57.0 ± 4.25 | 64.0 ± 3.15 [a] | 93.0 ± 4.45 [b] | 69.3 |
| ×1.0 | 79.0 ± 5.56 | 79.0 ± 6.66 | 78.0 ± 2.56 | 78.0 ± 4.23 | 14.0 ± 2.42 | 93.0 ± 4.12 [b] | 70.2 |
| ×1.5 | 83.0 ± 5.91 | 78.0 ± 7.77 | 64.0 ± 6.66 | 93.0 ± 4.12 [a,b] | 7.0 ± 3.34 | 79.0 ± 5.54 | 67.8 |
| ×2.0 | 86.0 ± 6.33 [a**] | 79.0 ± 5.52 | 71.0 ± 5.26 | 78.0 ± 2.16 | 14.0 ± 2.25 | 71.0 ± 5.12 | 66.5 |
| Factor B mean $LSD_{05}$ b = 6.27 | 82.3 | 73.3 | 69.3 | 76.5 | 24.8 | 84.0 | |
| | $LSD_{05}$ ab = 12.23 to compare private averages | | | | | | |
| | Average shoot length, cm | | | | | | $LSD_{05}$ a = 2.84 |
| ×0.5 | 7.9 ± 5.12 | 12.3 ± 4.82 [a] | 9.2 ± 5.62 | 9.9 ± 5.46 | 8.3 ± 5.50 [b] | 10.6 ± 5.42 [a] | 9.7 |
| ×1.0 | 6.7 ± 4.68 | 10.4 ± 4.84 [a] | 12.1 ± 4.61 [a] | 12.4 ± 4.25 [a] | 2.7 ± 0.25 | 10.9 ± 4.86 [a] | 9.2 |
| ×1.5 | 8.7 ± 5.11 | 15.0 ± 1.28 [a,b] | 13.6 ± 3.18 [a] | 13.4 ± 3.77 [a] | 2.6 ± 0.00 | 9.3 ± 6.06 | 10.4 |
| ×2.0 | 9.9 ± 5.41 [b] | 13.2 ± 4.08 [a,b] | 13.0 ± 4.99 [a] | 12.2 ± 4.71 | 2.5 ± 0.25 [a] | 8.7 ± 6.26 | 9.9 |
| Factor B mean $LSD_{05}$ b = 2.39 | 8.3 | 12.7 | 12.0 | 15.3, b | 4.0 | 9.9 | |
| | $LSD_{05}$ ab = 8.13 to compare private averages | | | | | | |
| | Sheet surface area, $cm^2$ | | | | | | $LSD_{05}$ a = 5.52 |
| ×0.5 | 17.9 ± 10.66 | 25.3 ± 9.51 [b] | 20.4 ± 9.96 | 21.8 ± 11.49 | 18.4 ± 11.99 [a] | 22.6 ± 11.46 [b] | 21.1 |
| ×1.0 | 16.5 ± 11.01 | 23.8 ± 9.23 [b] | 26.6 ± 11.78 [b] | 26.6 ± 7.76 [b] | 2.2 ± 0.65 | 25.0 ± 10.35 [b] | 20.1 |
| ×1.5 | 19.5 ± 8.64 | 31.6 ± 3.28 [a,b] | 29.8 ± 5.60 [b] | 28.7 ± 6.61 [b] | 4.5 ± 0.00 | 19.8 ± 13.24 | 22.3 |
| ×2.0 | 20.8 ± 8.28 | 28.8 ± 7.07 [b] | 27.7 ± 10.03 [b] | 26.9 ± 8.32 [b] | 1.9 ± 0.25 | 18.4 ± 12.69 | 20.8 |
| Factor B mean $LSD_{05}$ b = 4.65 | 18.7 | 27.4 | 26.1 | 26.0 | 6.8 | 21.5 | |
| | $LSD_{05}$ ab = 15.83 to compare private averages | | | | | | |
| | Average number of roots, pcs. | | | | | | $LSD_{05}$ a = 0.53 |
| ×0.5 | 3.5 ± 0.89 | 4.1 ± 1.05 [a,b] | 3.7 ± 0.94 | 3.8 ± 1.09 | 3.4 ± 1.07 | 4.0 ± 1.04 [b] | 3.8 |
| ×1.0 | 3.4 ± 0.88 | 4.1 ± 0.90 [a] | 4.1 ± 0.79 [a] | 4.5 ± 0.78 [a] | 2.0 ± 0.00 | 4.0 ± 0.78 [a] | 3.7 |
| ×1.5 | 3.6 ± 0.93 | 4.5 ± 0.50 [a,b] | 4.4 ± 0.68 [a,b] | 4.3 ± 0.72 [a,b] | 3.0 ± 0.00 | 3.6 ± 1.23 | 3.9 |
| ×2.0 | 3.7 ± 0.94 | 4.5 ± 0.66 [a,b] | 4.2 ± 0.98 [a,b] | 4.3 ± 0.75 [a,b] | 2.5 ± 0.50 | 3.4 ± 1.28 | 3.8 |
| Factor B mean $LSD_{05}$ b = 0.44 | 3.6 | 4.3 | 4.1 | 4.2 | 2.7 | 3.8 | |
| | $LSD_{05}$ ab = 1.51 to compare private averages | | | | | | |
| | Average root length, cm | | | | | | $LSD_{05}$ a = 1.08 |
| ×0.5 | 4.7 ± 1.97 | 6.1 ± 1.76 [a] | 5.1 ± 2.11 | 4.3 ± 1.72 | 4.5 ± 2.17 [b] | 5.5 ± 1.94 | 5.0 |
| ×1.0 | 4.4 ± 2.10 | 5.2 ± 1.32 | 6.0 ± 1.49 [a] | 5.8 ± 1.69 [a] | 2.0 ± 0.15 | 5.4 ± 1.96 | 4.8 |
| ×1.5 | 5.2 ± 2.10 | 7.2 ± 0.43 [a,b] | 6.3 ± 1.33 [a] | 5.9 ± 1.49 | 2.1 ± 0.00 | 4.9 ± 2.58 | 5.3 |
| ×2.0 | 5.5 ± 1.93 [b] | 6.5 ± 0.93 [b] | 6.1 ± 1.62 [a] | 5.9 ± 1.35 | 2.1 ± 0.10 | 4.5 ± 2.44 | 5.1 |
| Factor B mean $LSD_{05}$ b = 0.91 | 5.0 | 6.3 | 5.9 | 5.5 | 2.7 | 5.1 | |
| | $LSD_{05}$ ab = 3.09 to compare private averages | | | | | | |
| | Average total root length, cm | | | | | | $LSD_{05}$ a = 5.61 |
| ×0.5 | 17.0 ± 9.41 | 26.1 ± 11.04 [b] | 23.9 ± 11.67 [b] | 22.7 ± 10.21 [b] | 19.6 ± 11.83 [a] | 23.3 ± 11.75 [b] | 22.1 |
| ×1.0 | 17.2 ± 9.64 | 24.0 ± 9.32 [b] | 26.7 ± 9.34 [b] | 28.5 ± 9.12 [b] | 6.5 ± 0.50 | 25.9 ± 11.40 [b] | 21.5 |
| ×1.5 | 23.6 ± 9.61 [a] | 34.3 ± 3.49 [a] | 32.2 ± 6.37 | 29.9 ± 6.13 | 7.5 ± 0.00 | 20.1 ± 12.71 | 24.6 |
| ×2.0 | 23.5 ± 9.38 [a] | 30.0 ± 5.85 [a,b] | 27.9 ± 8.83 | 28.5 ± 9.26 [b] | 5.5 ± 0.50 | 20.2 ± 13.67 | 22.6 |
| Factor B mean $LSD_{05}$ b = 4.73 | 20.3 | 28.6 | 27.7 | 27.4 | 9.8 | 22.4 | |
| | $LSD_{05}$ ab = 16.08 to compare private averages | | | | | | |

The least significant difference (LSD) $p < 0.05$ was calculated by two-way analysis of variance. * the results are expressed as mean ± standard deviation (SD). ** "a, b, ab" are the differences between the averages when the recording is significant based on the comparison of the differences between the mean with LSD at a 5% significance level: "a" by factor "a" (experimental concentrations of iron chelates), "b" by factor "b" (iron chelates), and "ab" by the interaction of factors.

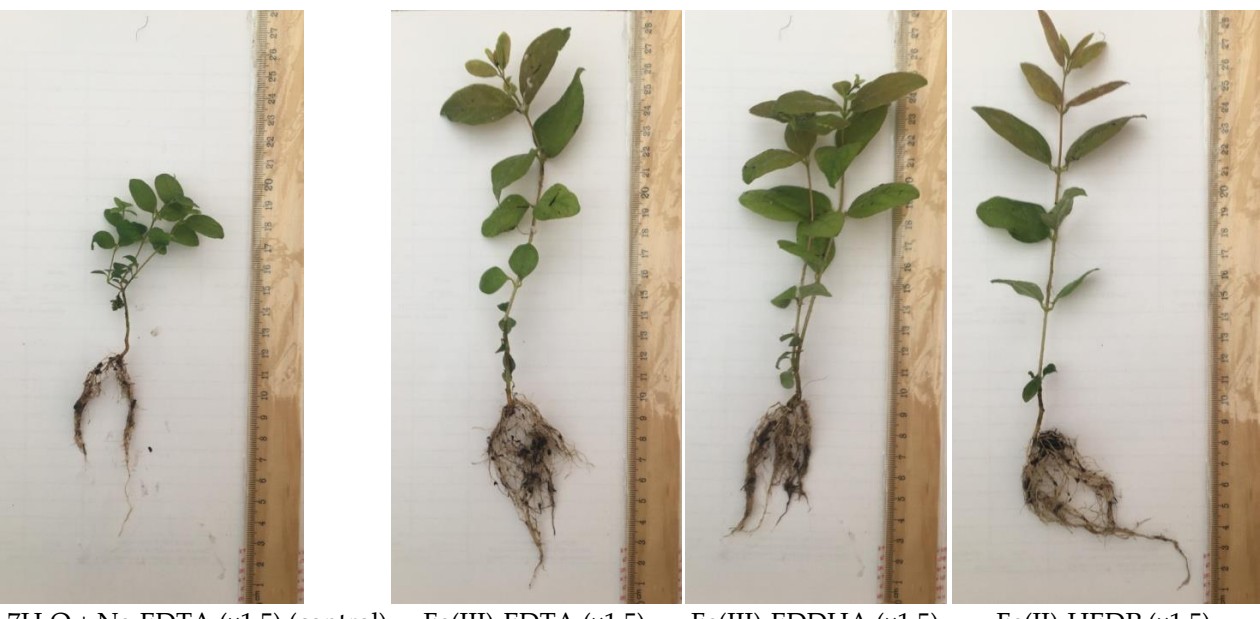

FeSO₄ × 7H₂O + Na₂EDTA (×1.5) (control)     Fe(III)-EDTA (×1.5)     Fe(III)-EDDHA (×1.5)     Fe(II)-HEDP (×1.5)

**Figure 5.** Blue honeysuckle "Lulia" (*Lonicera caerulea* var. *kamtschatica* Sevast.) plants at the end of the stage of acclimation to nonsterile conditions depending on the iron chelate type at an increased concentration (×1.5) in the MS nutrient medium at the stage of rooting.

The highest shoot and root growth indicators were observed in plants cultivated on nutrient media containing iron carboxyl chelates: Fe(III)-EDDHA, Fe(III)-EDTA, and Fe(III)-DTPA. The shoot length increased by 1.4–2.2 times, the total root length increased by 1.15–1.6 times, and the average root number increased by 22% compared to the control variant. The explants grown on a nutrient medium with bisphosphonate of divalent iron (Fe(II)-HEDP) also had increased growth and developmental indices, but only in two concentrations: reduced (×0.5) and standard (×1.0). However, as this study showed, such a positive effect of the nutrient medium modification on the growth and development of plants during acclimation to nonsterile conditions did not provide better plant rooting. Thus, the maximum plant survival rate (93%) was achieved in variants with organophosphorus (Fe(II)-HEDP) ligands applied at reduced (×0.5) and standard (×1.0) concentrations, and with Fe(III)-DTPA, at an increased concentration (×1.5) (Figure 3B). At the same time, the survival rate was increased by 18% compared with that of the control.

As well as at the stage of proliferation and rooting, the plant growth indices at the acclimation stage were in sharp contrast to those of the divalent and trivalent iron HEDP chelates treatments. Despite the absence of root formation in honeysuckle shoots when Fe(III)-HEDP was used at the rooting stage at standard and increased concentrations, the plants were planted in the acclimation stage without roots available to study their further development.

As expected, their survival rate was extremely low. Plants grown on nutrient medium with Fe(III)-HEDP had one comparable to the rest of the plants only at the reduced concentration (×0.5), 64.3%, which was, in any case, lower than the result of the control variant and 44% lower than the recorded survival rate while applying the divalent form with the same ligand. Fe(III)-HEDP at an increased concentration led to the sharp inhibition of plant growth and acclimation.

## 4. Discussion

In this research, we continued the cycle of studies on the extensive screening of iron compounds at different stages of the clonal micropropagation of fruit crops with the example of blue honeysuckle "Lulia" (*Lonicera caerulea* var. *kamtschatica* Sevast.). As in other previously published works, for example, with gooseberry plants, the obtained experimen-

tal results showed the functional importance of iron form in nutrient medium in terms of increasing the efficiency of multiplication technology and improving the characteristics of the obtained plants [37,38]. Using iron in the form of a complex (chelate) with one or another ligand improves the iron nutrition of plants, and this is an obvious fact for the vast majority of cultures. This fact has had numerous experimental confirmations, as highlighted with some references in the bibliography. In our research, by a direct comparison of the methods of obtaining metal complexonates and the preparation of the nutrient medium, we found that the medium did not meet the conditions of the chemical reaction of iron complexation with a carboxyl-containing ligand (EDTA). This idea challenged the availability of an iron complex in the right concentration in the nutrient medium and explained the physiological problems of explants.

However, as expected, the study revealed a specific susceptibility of honeysuckle plants to different forms of iron in a nutrient medium. At the stage of proliferation (stage S1) and rooting (stage S2), the plants clearly indicated a high sensitivity to carboxyl chelates such as Fe(III)-EDTA (multiplication) and Fe(III)-EDDHA (rooting). The application of the latter to the nutrient medium led to the improvement of all the biometric indices, as well as the shoot multiplication factor during proliferation and the rooting rate of honeysuckle microshoots during the rooting stage. Taking into account the results of the effect of chelated iron at the stage of honeysuckle rooting, we expected that the trend in general terms would remain in the subsequent stage—the acclimation to nonsterile conditions. A more powerful development of the root system and upper organs (stems and leaves) created the prerequisites for a successful acclimatization. The more interesting observations were the final results, according to which, despite the excellent biometric indicators of shoots grown by introducing, for example, Fe(III)-EDTA and Fe(III)-DTPA, their ability to survive and adapt was not very high. At some of the iron chelate concentrations in the modified nutrient media, the survival rate was significantly lower than that of the control variant, which negated the good external development of the explants.

At the same time, microplants grown on nutrient media modified with divalent iron bisphosphonate (Fe(II)-HEDP) were excellent at the final stage of acclimation (stage S3) and subsequent cultivation. The best results were obtained using reduced and standard proportions of concentrations according to the basic MS prescription, and at the rooting stage, the biometric performance of these explants was slightly worse than the best samples grown with Fe(III)-EDDHA, Fe(III)-EDTA and Fe(III)-DTPA. Thus, this study also confirmed our hypothesis that the use of phosphonate-containing ligands can stimulate the internal physiological potential of plant regeneration and increase the resistance to stressful acclimation conditions [41,42].

The reasons for the observed physiological reactions of the plants can be found in the chemical nature of the ligand, namely, the presence of phosphonic groups in the structure. On the one hand, it has long been known that many phosphonic acid derivatives have retardant properties, such as 2-chloroethylphosphonic acid [43]. Such compounds induce a metabolic response resulting in the triggering of enzymatic systems and other plant systems. Therefore, we were able to observe an obvious similar effect, the inhibition of shoot growth. However, hydroxyethylidene diphosphonic acid belongs to the class of compounds bisphosphonates with two phosphonic groups. This means that it is a complete structural analog of natural pyrophosphates, which in turn perform important metabolic functions at the cellular level. HEDP is able to interact with the ligands of membrane proteins in cells and integrate into these structures. Most aspects of the effect of HEDP on plant physiology and development remain unclear. We assume that the specificity of biological activity, despite some similarities with the known plant growth regulators based on phosphonic acid derivatives, is somewhat different. An important advantage is the ability to form chelate forms with divalent iron and other essential trace elements. The divalent metal transporter IRT1 allows a direct transfer of the element to the root epidermis, bypassing the plasma membrane proton release cycle and Fe(III) $\rightarrow$ Fe(II) via Fe(III)-chelate reductase. It should be noted that it is impossible to obtain chelate compounds of Fe(II) with

carboxyl-containing complexes because Fe(II) complexes are unstable and immediately transform into stable Fe(III) complexes. Thus, in one chemical compound, we have at the same time an excellent source of divalent iron available to plants and a growth regulator that helps increase plant stress resistance [43–45]. In our study, blue honeysuckle "Lulia" showed a much higher sensitivity to the valence form of iron bisphosphonate, with it showing compatibility only with $Fe^{2+}$. We believe that the reasons for this should be looked for in the genetic nature of the crop. On the other hand, the presence of phosphonic groups in the molecule structure provides a biostimulation effect and increases plant stress resistance [45,46].

Evidently, the most complete assessment can only be made only based on the combined analysis of the results of all stages: multiplication, rooting, and acclimation. It seems to us expedient to evaluate the indices of chlorophyll *a* and *b* for a clearer understanding of the mechanism of influence of various ligands in iron chelates. This indicator is very important for assessing biosynthesis and is more informative than just the iron content. However, we came to the conclusion that for a correct assessment, it was necessary to apply a different experimental research design scheme. In this study, we used a parallel scheme for the stages of proliferation and rhizogenesis. At the same time, earlier studies reliably showed that HEDP as a ligand exhibited retardant properties and aftereffects. For example, in the conducted studies in hydroponics conditions, a significant effect of the aftereffect of the use of $Fe^{3+}$ iron complexonate based on HEDP was demonstrated, in which the test plants after an environment with sufficient content $Fe^{3+}$ HEDP was moved to a nutrient solution without the iron chelate and, at the same time, the level of chlorophyll concentration in tissues and the intensity of photosynthesis did not decrease [47]. Thus, the measurement of chlorophyll indicators should be carried out dynamically, to record the change over time, but it is possible to do it correctly with a sequential scheme of experiments.

Taking into account the experimental data of each stage, we can conclude that the most suitable forms of iron for the clonal micropropagation of blue honeysuckle are carboxyl-containing Fe(III)-EDDHA and phosphonate-containing Fe(II)-HEDP. The efficiency of both complexants' influence was confirmed in a fairly wide range of initial concentrations: Fe(III)-EDDHA, from (×1.0) to (×2.0) and Fe(II)-HEDP from (×0.5) up to (×1.5). The use of carboxyl-containing iron chelates at the multiplication stage is preferred.

The variation in the survival rate and acclimation of microshoots at the stages of rooting and acclimation was significant. Consequently, in the in vitro and ex vitro stages, the survival, acclimation, development, and plant productivity should be ensured through a strict control of the honeysuckle's habitat based on the methodology of biosystem engineering [47,48]. In particular, a new protocol of in vitro and ex vitro cultivation, based on intrasoil-pulsed continuous–discrete humidification is necessary [49]. This will eliminate the intense hydrodynamic and static destruction and give a high stability to the substrate ex vitro. This will also open up the possibility to provide "training" of the plant at a relatively low soil moisture, improving the root system development ex vitro and productivity. The reduced thermodynamic potential of the soil solution reduces the mobility of heavy metal compounds and accelerates their passivation in the form of complexes and transfers them into insoluble forms of the soil's solid phase. Accordingly, heavy metals, to a lesser extent, come into the plant and improve product quality. In this case, due to the relatively high concentration of the soil solution, the electrochemical process of transferring the relatively easily mobile macroions and trace element ions of nutrients as well as the nanomaterials from the soil solution to the root system is intensified [50–53].

## 5. Conclusions

Studying the peculiarities of the multiplication, rooting, and acclimation of blue honeysuckle in in vitro culture practice, we confirmed the importance of establishing the correct iron source. Using the example of blue honeysuckle "Lulia", it was shown that blue honeysuckle exhibited a certain selectivity to the chelated form of iron. A modification of the nutrient media with chelated iron, to which the culture was sensitive, significantly

improved the physiological status of regenerated plants and increased the plant quality after acclimation. At the same time, the use of unsuitable iron sources or concentration values could lead to a decrease in the overall result compared with the use of the standard composition $FeSO_4 \times 7H_2O + Na_2EDTA$. In this regard, we recommend a more careful and thorough approach to the development of optimal technology for growing honeysuckle under in vitro conditions and to include the testing of divalent iron bisphosphonate Fe(II)-HEDP into the procedure selecting the optimal chelated form of iron, taking into account the prospect of a sustainable plant development ex vitro.

In addition, the divalent form of iron with a phosphonic ligand should be tested more widely on various crops, including tree crops. It is possible that many crops will show a very high susceptibility to the reagent, such as gooseberries and stone fruit rootstocks VTs-13 [37,38,53]. Therefore, it will soon be possible to improve the practice of the clonal propagation of plants and optimize the cultivation process.

The results of our study lead to the conclusion that the traditionally used indicators for assessing the growth and the physiological status of regenerant plants, such as shoot height, root length, and rooting rate, do not fully represent the quality and health of shoots at the stages of in vitro propagation. Plants subsequently have to undergo acclimation in nonsterile conditions and growing completion in minigreenhouses, during which stress resistance is more important.

**Author Contributions:** Conceptualization, A.G., S.A., E.N. and V.K.(Vadim Kirkach); methodology, S.A., E.N., V.K.(Vadim Kirkach) and E.R.; formal analysis, S.A., E.N. and A.K.; investigation, S.A., V.K.(Vadim Kirkach) and E.R.; resources, E.N., V.K.(Vadim Kirkach), N.T. and V.I.; data curation, A.G., M.P., A.S. and A.K.; writing—original draft preparation, S.A., E.N., A.R. and L.M.; writing—review and editing, A.G., V.K.(Valery Kalinitchenko), S.A., E.N. and L.M.; visualization, V.K.(Vadim Kirkach), M.P. and A.Z.; supervision, A.G., A.S., A.Z. and V.I.; project administration, A.G., V.K.(Valery Kalinitchenko) and A.R.; funding acquisition, E.N. and N.T. All authors have read and agreed to the published version of the manuscript.

**Funding:** The analytical research/research was done using equipment of NRC "Kurchatov Institute"—IREA Shared Knowledge Center under the project's financial support by the Russian Federation, represented by The Ministry of Science and Higher Education of the Russian Federation, agreement no. 075-15-2023-370 of 22 February 2023.

**Data Availability Statement:** Not available.

**Conflicts of Interest:** The authors declare no conflict of interest.

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
