# Peer review of "Preliminary Study: Micropropagation Using Five Types of Chelated Iron and the Subsequent Acclimation of Blue Honeysuckle (Lonicera caerulea var. kamtschatica Sevast.)"

_forests, doi:10.3390/f14040821_

Round 1

Reviewer 1 Report (Previous Reviewer 2)

The research has yielded interesting information regarding the influence and after-effects of some Fe salts in  micropropagation of honeysuckle. The manuscript may be published on some conditions. I hope it will be checked and corrected by a professional!!! translator before publication, as there are still spelling and stylistic mistakes and the language is still 'strange', 'overloaded' and convoluted.

 Some comments:

 The main two ones:

1) Table 1. The authors present detailed calculations of the Fe content for the tested experimental variants. However, the reader would be more interested in information about the final content of individual Fe salts in the media in order to  quickly reproduce the experiment. For example - what were the concentrations of Fe(II)-HEDP (not Fe only!) at studied doses? Therefore, at least an additional table containing information on the final Fe salt content should be very helpful.

Table 1. By the way - inconsistency: [mlL-1]  - why not [mL L-1] or [ml l-1]  !?

2) M&M  v.222 The experiments were performed in triplicate, at the stage of multiplication 5 plants cultures? per repetition and at the stage of rooting 15 plants shoots? per repetition.

There is still no precise information about the number of plants per 1 combination - 3x5? 15? 3x15? On the other hand, 15 cultures per combination is not enough number for experiment. Therefore, in the title and body of the manuscript it should be marked that it is a 'preliminary study'.

 Other ones:

v.36  during microplant accilmation??

v.50 Therefore, fruits, berries and  vegetables have a great importance for the healthy diet.  Berries are fruits! I've already pointed this out!

v.97 Scientists have researched 80 an effectiveness of adding auxins: IBA, IAA, 1-naphthaleneacetic acid (NAA) in the sstudy??. 

inconsistency: lack of IBA and IAA names? I've already pointed this out!

v.195 each cultural? vessel

v.230 medium coused? various explant reaction

Figure 2: Average shoot length  - lack of units [cm?]

Figure 3:  proportiopn ?

v.342 The higest? shoot and root growth indicators was observed in? plants cultivated on  nutrient

v. 365 As expected, their survival rate was extremely law??.

v.397 pro-portion?

v.445 In particular, a new protocol of ex? vitro? and ex? vitro? cultivation,

v.455 rel-atively?

v.465 acclimatization of  micrografts ?.  There were not micrografts! I've already pointed this out!

v.482 re - growth? in greenhouses,

Author Response

Reviewer 2 Report (Previous Reviewer 1)

Detailed comments on the amendments are provided in the attached pdf file.

Author Response

Reviewer 3 Report (New Reviewer)

The manuscript entitled “Micropropagation using five types of chelated iron and subsequent acclimation of blue honeysuckle (Lonicera caerulea var. kamtschatica Sevast)” aimed to study in detail the influence of different forms of iron at micropropagation stages of blue honeysuckle ‘Lulia’ (Lonicera caerulea var. kamtschatica Sevast.). The manuscript requires substantial revision in terms of results presentation and data analyses. Please see the following comments:

1- The manuscript should be polished deeply regarding language grammatical and structural revision. For example, L34: Tipe should be type

2- The introduction is long. It can be compacted by bringing important sections.

3- Material and methods don’t include all the detail of performed experiments. Please describe how you obtained and disinfected the explants. Additionally, please give codes to treatments.  

4- Unfortunately the provided graphs are not qualified and are not acceptable in the current form. It doesn’t meet the required standard for publication in a journal like Forest. First, if your experiment has been done with replications, so you should add SE values.

5- Higher quality images of regeneration and proliferation from the first steps are essential for research studies performed on plant tissue culture. I can not find such high-quality images in the present manuscript. So please add high quality from the first sign of regeneration to shoot development, rooting, and finally acclimatization and greenhouse cultivation

6- To have a better understanding of the Fe uptake and absorption it is better to quantify the Fe concentration within the leaf material of obtained plantlet. Why the respected authors didn’t do such analyses.

7- Finally only few characteristics have been evaluated which may not be enough to seed the effect of treatments. For instance, as Fe, directly and indirectly, involves in chlorophyll biosynthesis and structure one important factor could be the chlorophyll a, b, and total chlorophyll content of in vitro-raised plantlets.

With respect to the authors, the manuscript needs substantial revision and correction and it cannot go further evaluation providing doing the above-mentioned revisions.

Round 2

Reviewer 3 Report (New Reviewer)

The manuscript has almost improved based on the suggestions., however, I can not trace the changes as they have not been highlighted. Moreover, Fig 3 still needs to be improved and drawn in a better way. 

This manuscript is a resubmission of an earlier submission. The following is a list of the peer review reports and author responses from that submission.

Round 1

Reviewer 1 Report

The research topic and object are interesting and important, especially in the context of the growing interest in the fruits of this species. While natural Loniecera caerulea shrubs do occur in northern boreal forests, the research presented here is on cultivated varieties, so it is somewhat questionable that the manuscript has been submitted for publication in the journal Forests, particularly to the special issue of Forests, section Soil, entitled: Pollution, Haevy Metal, and Emerging Threats in Forest Soil. However, I leave this decision to the journal editors.

1) Staying with the name of the plant, I believe that it should be written in the manuscript as L. caerulea L. var. kamtschatica Sevast. unless the authors explain otherwise the botanical origin of the cultivar under study.

2) The authors should correct the name of the cultivar under study. They use it once as 'Luliya' and once as 'Lyulia', whereas the probably correct name of this Canadian cultivar is 'Lulia'. In addition, the notation 'cv. Luliya' is incorrect, because according to the international rules for naming cultivated varieties, it should look like this: 'Lulia' (without cv. and with single upper apostrohes). The name 'Luliya' proposed by the authors is probably a rusicism, but no changes to the original cultivar name are acceptable.

3. Line 31 (Abstract) - should be 5 types instead 4 types of chelated iron resources

4. Lines 38-40 - the sentence is missing a verb

5. Line 46-47 - Keywords are not correct: words in the title should not be repeated as keywords

6. Line 56 - berries are also fruits; so, it could be only "fruits and vegetables"

7. Line 59 - citing literature position 5 - This is absolutely too broad a reference. At the link provided, there is a FAO database of all the countries of the world and the cultivated areas or yields of many agricultural and horticultural crops. In my opinion, the reader cannot be asked to search for himself in the database for the necessary information to support the thesis that the importance of fruit and vegetable consumption is increasing. Some concrete examples should be given: a few countries and a few crops.

8. Line 75-78. The sentence should be corrected, as it talks about the development of blue honeysuckle production in different countries, and all citations refer to research on micropropagation, not the production. I suggest simply adding [...] production "and research". In addition, I do not see any justification in a scientific publication for emphasising that Belarus and Ukraine are former republics USSR. They are currently independent countries, and the cited Belarusian and Ukrainian research on blue honeysuckle micropropagation is from 2020 and 2018, respectively . Furthermore (line 77), I would like to ask if 'North America' means the USA? If so, please provide the correct name of the country. The reference No. 9 is from 1981, That is, from nearly 40 years ago, I suggest providing a more recent literature reference here.

9. Line 93 - repeating the full name of BAP here is a mistake. 

10. Line 114 and next lines in the paragraph, and last line of the chaper 5 as well. It seems to me that the use of the term 'in situ' here is incorrect. Can you clarify in what sense you are using it? 

11. Line 160-164 - information to delete - it's no place for the information about funding of the research. Such information are properly placed in separate section at the end of manuscript.

12. Line 168 - should be macro- and microelements instead micro- and macrosalt, and in this order (macro- first).

13. Line 170 - Please give the full name of ILA first. then abrreviation

14. Line 170 - mg/l to delete; agar-agar? It should be named with giving the name of producer as well

15. Table 1 - this table is completely unclear. Please explain why mg/L is given in the left column and percentages are given belowe? Why are successive concentrations identically labelled 'Decreased (x0.5)? Also note that previously the nutrient content of the medium was given in mg/l, in the table in mg/L. This should be standardised throughout the paper, and preferably the SI system should be used, i.e. the notation mg L-1. Where is the control? Also, the order of the listed iron chelates should follow the order in which they are presented in the results tables.

16. Line 178 - Please provide correct and contemporary units for the amount of light quanta (uM m-2 s-1) instead of the outdated nomenclature in lux. 

17. Table 2 (and further Fig. 1)- The title of the table should be improved. Please add the name of crop, please add the names of stages of evaluation. "Rooting ability" is not a measured parameter, but rooting ability was determined by the number and length of roots. According the figure 1, I see that you probably means  rooting ability as the degree of rooting in pecentage; silmilar "adaptation" = degree of acclimatization". This must be consistent and must be precisely defined. Because the degree of rooting is usually defined in terms of a scale of bonitation (e.g. 0-5 points/degrees), while here the percentage of rooted micro-cuttings is simply shown, the same for the percentage of acclimatised cuttings. In addition the parameters should be in the same order as in the tables of results. 

18. Statistical analysis - I think that data should be elaborated again by professional statitistical software (such a Statistica, IBM SPSS, StatGraphics), not by MS Excel 2007. You are written that standard deviations were calculated, but are not show anywhere afterwards. Statistical test for separating the means should be used. The control should be also add to the statistical analyses. 

19. Table 3 - Table is comletely unclear in meaning of statistical analysis - see above notes. In addition the types of chelates should be in the same order as in M&M (Table 1). What it means average length and total length? What is the difference? Why average length of shoots are provided in psc.??? 

20. Table - Line 253-255 - should be 4, not 3. The same notes as in the case of Table 3.

21. Line 258 - as a example of not clearly written the resukts. Which length of the shoots and roots do you mean? Average or Total? 

22. Reference 12. Latin name of the crop should be in Italic. "akademii navuk" sgould be as proper names in capital letters

23. Reference 14, line 405 - horts. should be delete

Only after corrections have been made to the statistical analysis of the results and the readability of the tables and graphs has been improved can the accuracy of the description, interpretation and discussion of the results and the proposed conclusions be thoroughly checked. 

English must be also improved.

Reviewer 2 Report

The authors present the results of application of five types of iron chelates  in the last stages of honeysuckle  micropropagation. The clone-specific reaction onto different medium ingredients is well-known phenomena and inconvenience in propagation of many plant species. Therefore the aim of  study, despite of existence of many information about FeNaEDTA and FeEDDHA influence on in vitro cultures is rationale from practical point of view to optimise micropagation of specific clones. The authors describe also the effects of other, less frequently used chelated Fe forms (DTPA, HEDP) what increases the novelty value of presented study. Nevertheless, a question arises why  they didn’t examine these chemicals in the first stages of micropropagation (initiation, proliferation)? Lack of such results strongly decreases ‘practical’ value of research. There are not any results of analyses of plants physiology thus the value of manuscript for basic science is poor. It has also many disadvantages which preclude its acceptance in this form.

Some other comments:

 Abstract: “Each type of iron chelate was tested at 4 concentrations: standard, decreased by 2 and increased by 1.5 and 2 times in the basic nutrient medium Murashige and Skoog (MS).” -  The more precise data should be given (mg/l? mM?). 

 Keywords: honeysuckle; micropropagation; acclimatization; iron; chelate; hydroxyethylidene diphosphonic acid; Biogeosystem Technique (?)  Two words are the same as in title.

 v.93  hidiazuron ?

  Do simplify “honeysuckle of cv. Lyulia” onto honeysuckle ‘Lyulia’

 Introduction. It seems to be too long. Consider shortening especially the first 3 paragraphs.

 Materials and Methods

Plant Material – lack information about in vitro cultures – when were they established, how many subcultures, the composition of medium used in proliferation stage, etc.

v.168  “microsalts, macrosalts” – micronutrients, macronutrients?

v.170 ILA?  -   chemical name ?

“The nutrient medium, previously poured into culture vessels” - What were the volumes of vessel and medium?  How many explants per vessel?

 v.178 “16-hour  photoperiod” -  16h/8h (d/n)  photoperiod?

v.177 “micro shoots 2-3 knots long”  - microshoots 2-3 node?

Was the peat/perlite mixture used in plantlet acclimation enriched with fertilizer? What was its dose and composition?

 Table 1.  unclear data – First column:  Complex, mg/L  below  %??? 

 What were the molecular weights of the tested Fe chelates? What was the Fe content in the molecule? Was the influence of Fe chelates compared at the same molar or weight concentrations?

 Statistical analysis

 How many explants and plantlets represented each combination? Have the experiments been repeated?

The statistical analyses seems to be too poor. Why the authors did not LSD multiple mean separation test? They should be helpful in results interpretation. Data presented as percentage ( number of rooted shoots, etc.) should be transformed or subjected to test on difference among proportions.

 v.197 “two-factor analysis of variance, calculations of the  arithmetic mean, standard deviation, coefficient of variation”. Some data (SD, CV) is not presented in Results!? What was the second factor in ANOVA?

Results

Did Authors observe any symptoms of physiological disorders?

 Table 3. Unclear (the same?) LSD letters/signs

Least significant difference P < 0.05 a?

Least significant difference P < 0.05 b?

Least significant difference P < 0.05 ab?

What is the difference among a , ab and b?

 v.207 “…tendency of plant response to the introduction of various  forms of iron…” - tendency of plant response to various forms of iron?

v.217 “Fe(III)-EDDHA at an increased concentration (×1.5) and Fe(III)-DTPA at all tested concentrations, the explants developed 218 1.47-2.0 times more sprouts, but they were shorter” – in comparison with?

 Consider using spider charts for quick and comprehensive visualization of the effects of test compounds. And cluster analysis…

 Introduction and Discussion.

I think that considerations and conclusions about plant in vitro culture response and “Biogeosystem Technique” go too far, are poorly justified, therefore are not necessary.

 Summarizing, despite of some interesting results I recommend to reject the manuscript in such a form. However it may be reconsidered after great improving and extension with additional results and information (as a completely new manuscript).

Round 2

Reviewer 1 Report

Dear Authors,

As a general conclusion, I must say that the article has been improved especially with regard to the presentation of the results. However, at many points the Authors have not addressed my comments, despite stating that they have taken the comments into account. I give selected examples below.

1) The Authors have decided - to use the name Lonicera caerulea L. in the title and parts of text - despite my suggestion that it should be Lonicera caerulea var. kamtschatica Sevast. Please justify your decision. On the other hand, in the Material and Methods chapter, in the Plant Material subsection, they use the suggested name L. caerulea var. kamtschatica Sevast. Please be definitive about which botanical name will be used in the paper and be consequent in doing so.

2) Keywords are still not fully correct: Authors still left two words from the Title of manuscript (honeysuckle and iron).

3) My previous request to remove the word 'berries' in the second sentence of the Introduction chapter, because berries are also fruits, was not complied with, although the Authors stated that they had made the correction.

4) My previous remark No. 7 concerning citing literature position 5 (faostat) is left completely without reaction. The updated link means nothing, because my comment was about something completely different - please re-read it carefully and refer to it. Simply refreshing the link changes nothing, because the problem I pointed out remains - the same page opens and the Authors expect readers to search for the data themselves. 

5) No response to my previous comment No. 11, although the authors state that they have removed information about research funding from the Materials and Methods chapter. In the text of the manuscript this information remains.

As a reviewer, I expect the Authors to respond to my questions and suggestions and comments. They do not have to agree with them, but they should respond to them. I get the impression that my previous work was in vain. I would also add that the text has many more editorial errors (extra spaces, missing spaces, extra full stops) than the first version of the manuscript.

In this form the article cannot, in my opinion, be published. I am willing to review it again if the authors reliably address the comments from the first review.

Reviewer 2 Report

It seems the authors have made some efforts to improve the manuscript. However, they probably acted in a hurry and left a lot (!!!) of different mistakes and bugs, which makes the „revised”(?) version unsatisfactory L  In my private opinion (I am not an expert in the English language) the manuscript requires 'an extensive editing of English language and style'. The authors use terms in improper meanings (for example: in situ, micrografing, microgreens, micropods, engraftment, microseedling, …). There are so many shortcomings that I can't list them all. However, I present some of them: 

Keywords: Still some the same words as in title. Other - 'substrate and soil water regime' - inappropriate. What about: 'in vitro', 'propagation'? 

Introduction. "Its advantages have contribute to a development of honeysuckle production in Belarus, Ukraine , in the USA and Canada, China, as well as in a number of Eastern European countries (Czech Republic, Slovakia, Romania, Poland, Croatia)." Hm, possible that from the russian point of view these are Eastern European countries. However, most of them are countries of central Europe, whereas Belarus and Ukraine are landlocked in Eastern Europe. What about: "in a number of European countries (Belarus, Croatia, Czechia, Poland, Romania, Slovakia, Ukraine)"? 

v.93  tidiazuron > thidiazuron 

v.97 „When scientist focus on selection of hormonal components for medium , a problem of using vital nutrients, such as iron, can be.” – style!  L 

v.>172 „In the present work, taking into account the great production potential of honeysuckle in the sector of healthy diet and consequently , the growing interest in mass production of this berry, we aimed to study l in detail the influence of different forms of iron at the stages of multiplication and rhizogenesis of clonal micropropagation of honeysuckle ‘Lulia’, and also following ac-climatization and rearing (?) in non-sterile conditions.” – style!  L Do simplify this sentence and divide into 2-3 ones. Micropropagation is usually a method of clonal propagation…  

Thus, we sought to determine the effectiveness of this technique for the entire technological cycle of micrograft(?) cultivation in the perspective of ensuring successful organogenesis (?) of plants in situ (?).  Honeysucle is usually propagated by cuttings or in vitro, not by grafting or micrografting! 

What do you mean by: ‘in situ’? In the field (conditions)?, organogenesis (?) – performance? 

M&M 

v.>186  Studies were carried out on a variety of  - no need for italics 

Use ‘cultivar’ or ‘clone’ instead ‘variety’ 

v.>224 „At the stage of multiplication, microplants(?) honeysuckle” -microplants > explants? What kind of explants were used: shoot tips, nodal ones? What size? 

v.>229. „The  nutrient medium was poured into glass culture vessels with a volume of 200 ml, while  each vessel was filled with 30 ml of the medium. Microcutting(?) was carried out by 5  pieces(?) in each culture vessel. The duration of subculturing at the stage of multiplication  was 60 days. The age of the culture is 1 year (6th passage).  Style! L Why don't you use the wording/schemes/terms used in articles devoted to micropropagation? 

v.>241 „Micrografting(???) was carried out in glass culture vessels with a volume of 200 ml, 15 pieces  each, the duration of subculturing was 60 days”. I think you did not use micrografts but microcuttings / explants! 

The nutrient medium was poured into the vessels, then sterilized in the autoclave at a temperature of 120°C and a pressure of 0.1 MPa for 20 min. In the laminar box, 5 micro shoots with a length of 2-3 node . . Cultures then incubated in the light room at the intensity of illumination of 46.25 uM m-2 s-1, 16h/8h (d/n) photoperiod, and a temperature of 20-22°C. On 45 and 60 days of subculture , morphometric indicators of microbes were recorded.” - Style! L Why don't you use the wording/schemes used in articles devoted to micropropagation?  

intensity of illumination? – not PPFD? 

v. 288 – „morphometric indicators of microbes”? - microbes??? 

v.> 294 „Nitrogen (NH4 + N03) 100 mg/l, Phosphorus (P2O5) 90 mg/l; Potassium (K2O) 120 mg/l;” - Inconsistent notation: NH4 – numbers in normal fonts, P2O5 – numbers in subscript 

v.298 „Previkur” -   Previcure? 

v.>302 „During the adaptation period, microplants were subjected to  preventive quenching by short daily ventilation with extended duration” – What do you mean by ‘preventive quenching’? 

Table 1 

first column 'Complex' - remove 'mg/l' close to chemical symbols   

put unit symbols in square brackets 

Decreased (2x)” - not 'increased'? 

v.>314 Statistical analysis 

Experiments were performed in triplicate, 50 plants per repetition. Repetition of experiments was three times, 25 plants per repetition.” UNCLEAR ! Then 50 or 25 plants!? Did you test iron chelates in one or three subcultures at multiplication stage and one or three subsequent rooting/aclimatization stages? Why don’t you present the results of each subculture separately? The reaction of cultures might be different because of habituation. In v.>241 you have written: „Micrografting was carried out in glass culture vessels with a volume of 200 ml, 15 pieces  each”. 25 and 50 is not divided by 15. How do you explain it?, 

To the best of my knowledge, STATISTICA indicates significant differences among the means but does not give the LSD values. How were they calculated? 

Results.and Discussion 

I suggested to use spider charts to compare comprehensively the influence of studied chelates on different traits, not only one (multiplication ratio or shoot length). 

Table 3. – still unclear! Many mistakes! 

For example: As factor A is the concentration of chelates – the mean for control is 86.8% (rooting), not 79% (mean for factor B ). Confused definitions of factor A and B? 

Where is the value for LSD for factor A? What do „HCP05a, НСР 05 ab” mean? 

фактор b – cyrillic font!? 

„Fe(III)-EDDHA+ SD” - What does  „+ SD" mean”? 

What do you mean by: „private averages” ?

80.0±2.47а88” - What does „а88” mean? 

Table 4. – similar questions as for Table 3. 

Additionally: Title of third column  ‘Factor average A+ SD’ seems to be unproper! 

„FeSO4×7H2O + Na2EDTA+ SD* (control)”, „×1.0(control)” - Two different controls? 

Sheet surface area, cm2” - What do you mean by: „Sheet Surface”? 

Figure 3 - Lack of rooted shoots (Fig.3A) and about 8-12% of acclimatised microplants (Fig.3B) for Fe(III)-HEDP (1-2x)! How do you explain it? 

v.324 „Testing of different iron complexes in nutrient medium on the proliferation process , very different response of explants both to the complexes themselves and to their concentration.” Style! L incomplete sentence 

v.337 „The values of the multiplication factor were close to the control and in some cases complex lower.” Style! L 

v.>382 „In spite of fairly good ability of this culture to spontaneous rooting on standard nutrient medium, introduction of chelated forms of iron had additional stimulating effect up to 100% rooting of all specimens (Figure 1 (A)). -  wrong figure number; specimens, not shoots/explants? 

v.>475 „Increasing the concentration of Fe(III)-HEDP led to a dramatic increase in inhibition of plant growth and adaptation(?) adaptation”. What about simplified version: Fe(III)-HEDP in increased dose/concentration led to a dramatic inhibition of plant growth and acclimatization? 

v.484 „regenerant plants” – regenerated plants, obtained plants, micropropagated plants?  

v.508 What do you mean by:  „micropods”? 

v. 501 „Having available preliminary data on the effect of  chelated forms of iron on rhizogenesis of honeysuckle, we could assume that the trend  in general terms would remain in the subsequent stage  - Do you make assumptions based on preliminary data !? 

v.>447 „The shoot length increased by 1.4 – 2.2 times, and the total length of the roots length by 1.15-1.6 times, the number of roots by an average of 22%”.- IN COMPARISON WITH !?!? 

Do you see the difference between ‘adaptation’ and ‘acclimatization’ terms? 

‘tree crops’ – ‘woody crops? 

v.643 – ‘Microseedlings’ -  You did not propagated plants by seeds in vitro. Thus ‘microseedling’ term is improper.  

Summarizing, the authors had enough time to improve manuscript. Unfortunately it contains too many mistakes and is written in convoluted, overloaded and obscure language. It seems that the authors present the results of one multiplication subculture and one rooting/acclimatization stage. Thus the results are preliminary. Therefore, despite of some interesting results I recommend to reject the manuscript. 

Round 3

Reviewer 2 Report

I think that manuscript was corrected, even improved,  but some changes are improper. In my private opinion (I am not an expert in the English language) the manuscript requires 'an extensive editing of English language and style' including also biological/scientific terminology. It is still written in strange English.

 Some comments:

 Lonicera caerulea var. kamtschatica Sevast. [syn. Lonicera kamtschatica (Sevast.) Pojark] is botanical variety (var.) which contains some cultivars (cultivated varieties), like 'Lulia'. Therefore I think that formulation " Lonicera caerulea L. cultivar kamtschatica " is improper and should be replaced by " Lonicera caerulea var. kamtschatica Sevast." or "Lonicera kamtschatica (Sevast.) Pojark".

 A single apostrophe: " 'name' " is used to denote a cultivar name. Thus  ‘Lulia’ = cv Lulia = cultivar Lulia. You do not have to write: "… the honeysuckle of ‘Lulia’ cultivar (Lonicera caerulea L. cultivar kamtschatica Sevast)" or even: "honeysuckle of ‘Lulia’ cultivar".  "Blue honeysuckle ‘Lulia’ " is enough in the text !

 Because of some confusion with "acclimatisation" and "adaptation" terms the term "acclimation" is proposed as the name of the last stage of micropropagation.

 phytomers not metamers  !  (terms for plants and earthworms, respectively)

 uterine!?!?! plant -  - What do you mean by:  "uterine" ?

 I insist to use the units of PPFD thus: 46.25 μM m-2 s-1 PPFD

 There are still some shortcomings :

various iron form! treatments / rootind! rate / concentration proportiopn!  / reсordered!?  (not recorded?), 2 mlL-1 (not:  2ml / l, 2mL/L  or cm3 dm-3)

 The manuscript still needs improvement.  :/
